# Leveraging technology to probe mechanisms of psychopathology: A proof of concept study of inhibitory control

Elise M. Cardinale[1]*, Jennifer M. Meigs[1], Simone P. Haller[2], Kenny Fling[2],
Urmi Pandya[3], Olivia Siegal[4], Anjali Poe[2], Shannon Shaughnessy[5], Christian Zapp[6],
Jessica L. Bezek[7], Kyunghun Lee[2], Parmis Khosravi[2], Ramaris German[2],
David C. Jangraw[8], Lauren M. Henry[2], Meghan E. Byrne[2], Katharina Kircanski[2],
Ellen Leibenluft[2], Reut Naim[9], Daniel S. Pine[2], Melissa A. Brotman[2]

**1** Department of Psychology, The Catholic University of America, Columbia, Washington, United States of America, **2** Emotion & Development Branch, National Institute of Mental Health, Bethesda, Maryland, United States of America, **3** Department of Psychology, The University of Washington, Seattle, Washington, United States of America, **4** Department of Psychology, Yale University, New Haven, Connecticut, United States of America, **5** Department of Psychology, University of Miami, Miami, Florida, United States of America, **6** Kaiser Permanente Bernard J Tyson School of Medicine, Pasadena, California, United States of America, **7** Department of Psychology, University of Michigan, Ann Arbor, Michigan, United States of America, **8** College of Electrical and Biomedical Engineering, University of Vermont, Burlington, Vermont, United States of America, **9** School of Psychological Sciences, Tel Aviv University, Tel Aviv, Israel

* cardinale@cua.edu

## Abstract

### Objective

Quantifying relevant behavioral mechanisms has relied on rigorous, time-consuming tools restricted to laboratory settings and inaccessible to the clinical community. Advances in technology provide an opportunity to develop more accessible platforms. Here, we developed CALM-IT, a novel mobile-application to experimentally assess inhibitory control in vivo

### Method

In a transdiagnostic sample of 200 youth aged 8–20, we (i) apply knowledge from canonical inhibitory control tasks in the methodological design of the mobile application, (ii) establish feasibility and engagement with CALM-IT, (iii) assess test-retest reliability of CALM-IT, (iv) investigate the convergent validity of CALM-IT with behavioral and neural responses to laboratory-based tasks, and (v) probe clinical relevance via associations with clinical symptoms.

### Results

First, we provide evidence that our novel inhibitory control mobile application, CALM-IT, was accessible, feasible, and engaging. Second, we found performance was reliable

**Data availability statement:** All data files for participants who consented to data sharing are publicly available at Open Neuro accessible via the following link: https://openneuro.org/datasets/ds005166/versions/1.0.0

**Funding:** This research was supported by the NIMH Intramural Research Program (ZIAMH002781), conducted under NIH Clinical Study Protocols 15-M-0182 (ClinicalTrials.gov identifier: NCT02531893), 02-M-0021 (NCT00025935), 00-M-0198 (NCT00006177), and 01-M-0192 (NCT00018057). The funders had no role in study design, data collection and analysis, decision to publish, or preparation of the manuscript.

**Competing interests:** The authors have declared that no competing interests exist.

over time. Third, we found CALM-IT performance was associated with established measures of inhibitory control and activation in the bilateral inferior frontal gyrus. Associations with brain but not behavior survived after controlling for age. Finally, we found evidence linking impaired CALM-IT performance to increased levels of co-occurring anxiety, irritability, and attention deficit hyperactivity disorder (ADHD) symptoms.

## Conclusion

Validation of this neuroscience-informed mobile application represents a critical first step in bridging precise, mechanism-driven research and community-based assessment of childhood psychopathology. The present work lays the groundwork for future research that could provide researchers and clinicians with a multifaceted tool to measure clinically-relevant behaviors in an engaging and accessible manner.

## Introduction

Identifying behavioral deficits associated with psychopathology facilitates a mechanism-based understanding, [1,2] but there has been limited integration of behavioral assessments in clinical practice [3,4]. Challenges remain regarding the availability and accessibility of evidence-based instruments. Quantifying behavioral mechanisms traditionally has relied on rigorous, time-consuming tools largely restricted to laboratory settings [5] and generally inaccessible to practitioners. The proliferation of mobile tools, coupled with advances in technology, provide an opportunity for clinicians to use more accessible platforms [6]. However, most such research focuses on repeated measurements of symptoms rather than behavioral performance on tasks [7]. Here, we provide a framework for leveraging mobile technology to bridge granular probes of behavioral mechanisms with more accessible mobile metrics.

Specifically, we describe the novel mobile application to measure inhibitory control, "CALM-IT." We designed CALM-IT to build off the strengths of traditional laboratory-based cognitive control tasks while addressing issues with engagement, accessibility, and cost-effectiveness of these often laborious and time-intensive experimental tasks particularly within developmental samples [5]. To do this, we leveraged knowledge from canonical inhibitory control laboratory tasks and recent advances in the gamification of inhibitory control assessment. By using a game-based interface that can be completed using mobile-devices, CALM-IT was designed to extend traditional laboratory-based cognitive control tasks into an engaging, accessible mobile assessment of inhibitory control. While gamification has proven an effective technique for increases engagement while preserving validity of performance, findings are mixed [8–10]. Therefore, it is critical that these novel apps undergo systematic evaluation. As such, we provide preliminary data demonstrating feasibility, accessibility and preliminary validity of this mobile application [11]. Using the development of "CALM-IT" as a proof of concept, we present a roadmap for the development of novel mobile-based behavioral assessments to further better understanding of mechanisms of psychopathology.

Inhibitory control represents a particularly strong candidate as a behavioral target as it represents a well-validated construct. As a well-validated neurocognitive mechanism, inhibitory control encompasses an individual's capacity to withstand or regulate impulses and suppress automatic, or prepotent, behavioral reactions [12]. Extensive research investigates inhibitory control spanning species, [13–15] stages of development, [16–18] and research methodologies [19–22].This research has generated several laboratory tasks that share neural circuitry, involving ventral prefrontal cortex and associated regions [13,23].

Inhibitory control has been implicated as a potential mechanism underlying childhood psychopathology. Childhood psychopathology, including mood and attentional difficulties, has been linked to difficulty inhibiting responses and atypical neural responses during the inhibition of motor responses [24–27]. Our pilot work with "CALM-IT" has a focus on addressing anxiety, irritability, and ADHD symptoms, as these are common, impairing, and tend to co-occur across a spectrum of childhood psychopathology [28–30]. Furthermore, atypical cognitive control associated with anxiety,[31,32] irritability, [33,34] and ADHD [16,35] symptoms has been posited as a potential mechanism for the comorbid presentation of these symptoms

The current work provides a roadmap for leveraging mobile technology in the development of clinically-relevant tools, to bridge the granularity of neuroscience-informed assessments of behavior into an accessible and disseminable format (Fig 1). We present the roadmap for leveraging technology to develop an accessible and potentially clinically relevant tool (Fig 2): (i) apply knowledge from canonical laboratory-based behavioral tasks to inform the design of the mobile application, (ii) establish feasibility and engagement with the mobile application, (iii) demonstrate the reliability of the mobile application-derived behaviors, (iv) investigate the validity of application-derived behaviors through associations with behavioral and neural responses during canonical laboratory-based tasks, (v) probe clinical relevance and associations between application-based behaviors and clinical symptoms.

In this paper, we used the development of CALM-IT as a proof of concept for this roadmap, providing a concrete example of implementing each of the roadmap's steps. First, we applied knowledge from two canonical inhibitory control tasks, Go/No-Go [36] and Stop Signal [37] Delay Paradigms, to inform the methodological design of CALM-IT. Second, we assessed feasibility and engagement with CALM-IT and hypothesized that we would observe high rates of task completion and performance accuracy. Third, we assessed test-retest reliability of CALM-IT across two sessions spaced approximately 1-week apart and hypothesized that inhibitory control performance as measured by CALM-IT would be reliable over time. Fourth, we assessed convergent validity of CALM-IT against a latent variable of inhibitory control behavior estimated from participants' task performance across four laboratory tasks and neural response to cognitive conflict using the Eriksen Flanker Task. We hypothesized that CALM-IT performance would be associated with both behavioral and neural lab-based metrics of inhibitory control. Finally, we assessed clinical relevance through associations of CALM-IT performance and parent- and child-reported symptoms of anxiety, irritability, and ADHD. We anticipated deficits in CALM-IT inhibitory control performance would be associated with increased levels of psychopathology.

## Methods & materials

### Participants

Two hundred youth aged 8–20 (M[SD]=13.59[2.91]) were recruited to participate in research at the National Institute of Mental Health from June 19th 2019 through June 28th 2023. Participants were recruited as part of several larger protocols clinically characterizing patients with a primary anxiety disorder, disruptive mood dysregulation disorder (DMDD), oppositional defiant disorder, or ADHD, as well as typically developing youth with no psychiatric diagnosis (Table 1). All diagnoses were confirmed using the Kiddie Schedule for Affective Disorders and Schizophrenia for School-Age Children-Present and Lifetime version (KSADS-PL) [38] and reviewed by Board-Certified psychiatrists or psychologists. Exclusionary criteria included IQ<70, neurological disorders, diagnosis of bipolar disorder or autism spectrum disorder, past and/or current

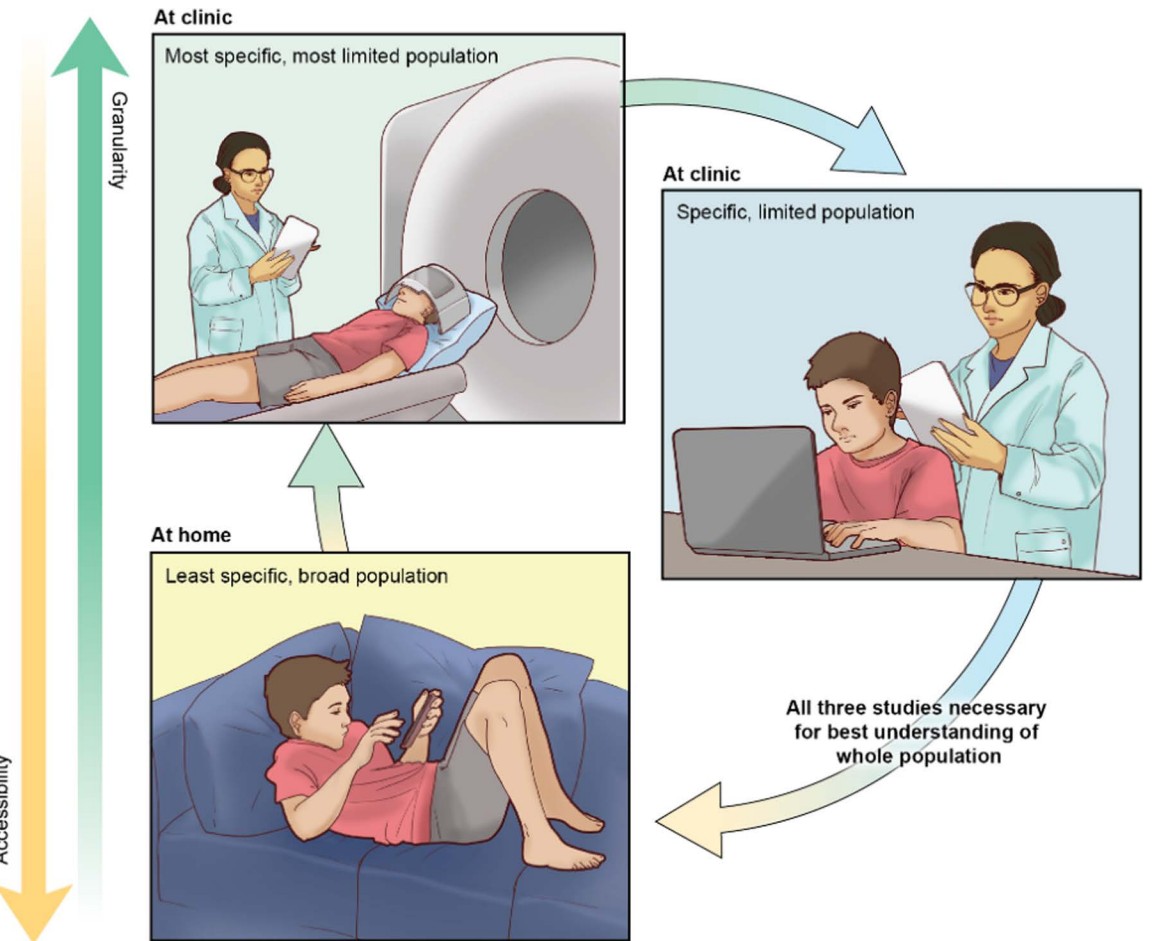

**Fig 1. Leveraging mobile-technology in the development of clinically-relevant tools to bridge the granularity of neuroscience-informed assessments of behavior and the accessibility and easily disseminated format of mobile devices.**

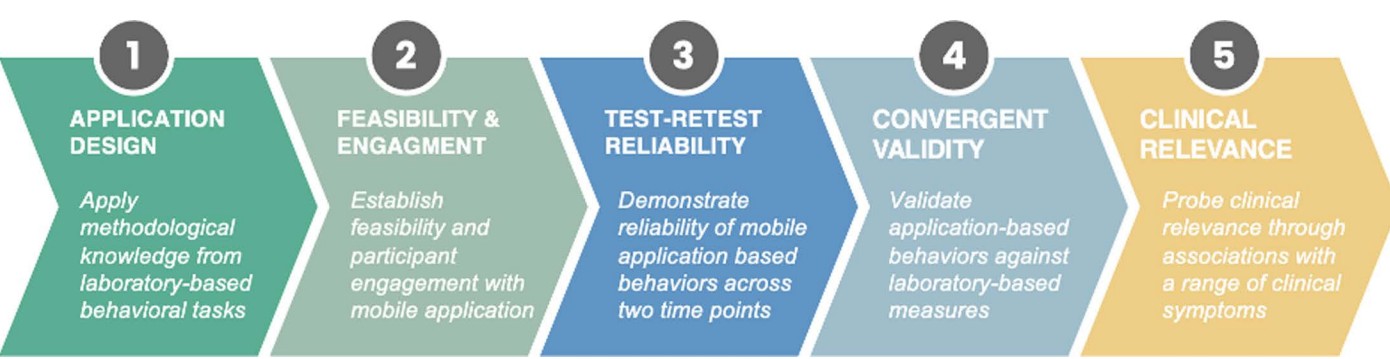

**Fig 2. Roadmap for the development and validation of accessible and clinically-relevant tools for experimentally assessing behavior in vivo.**

**Table 1. Sample Characteristics.**

| Variable | M (SD) or No. (%) |
| --- | --- |
| **Demographic Variables** | |
| Age (years) | 13.59 (2.91) |
| Sex (% female) | 91 (45.5%) |
| IQ[a] | 113.12 (13.25) |
| Race | |
| Caucasian | 141 (70.5%) |
| Multiple Races | 27 (13.5%) |
| Black or African American | 17 (8.5%) |
| Asian | 8 (4.0%) |
| Unknown | 4 (2.0%) |
| American Indian or Alaskan Native | 2 (1.0%) |
| Other | 1 (0.5%) |
| Ethnicity | |
| Unknown | 3 (1.5%) |
| Not Hispanic or Latino | 179 (89.5%) |
| Hispanic or Latino | 18 (9.0%) |
| **Clinical Measures** | |
| Research Group[c] | |
| ADHD | 56 (28.0%) |
| Anxiety Disorder | 35 (17.5% |
| DMDD | 31 (15.5%) |
| Sub-DMDD[d] | 26 (13.0%) |
| No Diagnosis | 52 (26.0%) |
| Medications[e] | |
| None | 178 (89.0%) |
| SSRI | 11 (5.5%) |
| Stimulant | 11 (5.5%) |
| SGA | 0 (0.0%) |
| AED | 1 (0.5% |
| Other | 9 (4.5%) |

*Note.* ADHD – Attention Deficit Hyperactivity Disorder, DMDD – Disruptive Mood Dysregulation Disorder, SSRI – Selective Serotonin Reuptake Inhibitor, SGA – Second-Generation Antipsychotic, AED – Antiepileptic Drugs.

[a]38 participants were missing IQ scores.

[b]Socioeconomic status (SES) based on current occupation and highest level education of the participant's parents.

[c]Research Group refers to the diagnosis for which the participant was referred. Participants could have multiple diagnoses in addition to their primary diagnosis.

[d]Youth with sub-DMDD were required to exhibit temper outbursts at least once per month, irritable mood at least one day per week for most of the day, and irritability-related impairment in at least one setting (home, school, peers); all other criteria for sub-DMDD were the same as those for DMDD.

[e]Percentage of participants taking each medication type is reported. Participants could be taking more than one type of medication.

posttraumatic stress disorder, schizophrenia, or substance use within the past three months. Prior to participation, parents provided written informed consent and youth provided written assent. Families received monetary compensation for their participation. All materials and procedures were approved by the National Institute of Mental Health Institutional Review Board.

## Procedures

All participants were sent instructions on how to download CALM-IT on their mobile device, or they were temporarily provided with a mobile device with CALM-IT downloaded, allowing the mobile application CALM-IT to be completed at a time and place of the family's choosing.

To assess test-retest reliability of app-based behavioral performance, a subset of the sample (N = 133) agreed to play CALM-IT for a second time approximately 1 week later ($M_{days}$[SD]=10.95[9.81]). To assess the convergent validity of app-based task performance, we extracted data for 118 participants who separately had completed in-laboratory testing on four canonical inhibitory control tasks (days between tasks: $M[SD]_{days}$ =470[420]). Finally, data for a subsample of 77 youth had completed a cognitive conflict fMRI scan as part of a separate protocol was also extracted (days between tasks: $M_{days}$[SD]=133[507]).

## Measures

**Inhibitory control mobile application.** In collaboration with an industry partner, we developed a mobile application, CALM-IT, which assesses motor inhibition using a gamified interface. During gameplay, participants explore ten galaxies, while moving through outer space as an astronaut. Within each galaxy, or level, participants' objective is to destroy space objects by swiping the screen with their finger to hit each target, while at the same time avoiding hitting stars. The space objects are depicted as red, purple, or blue objects, while stars appear in yellow (Fig 3, Panel A & B). Participants earn two points for every target hit and lose one points for each star they hit. When a star is hit, an explosion animation appears on the screen and a visual indicator of points lost appears next to the running point tally (Fig 3, Panel C). After completing each level, a cumulative score screen is presented to participants (Fig 3, Panel D).

CALM-IT consists of 10 separate galaxies (e.g., levels), each includes 13 stars and 39 targets. Each level takes about 1 minute to complete, for approximately 10 minutes of gameplay in total. The first five galaxies were modeled after the Go/No-Go paradigm [36] with targets as the "go-stimuli" and stars as the "no-go stimuli". The last five galaxies were modeled after the Stop Signal Delay paradigm [37]. In these levels, 25% of the targets turn into stars at a random time within two seconds of appearing on the screen. Importantly, unlike the standard Stop Signal Delay task, the time delay between when the targets-turned-stars appear on the screen and turn into a star is not tied to participant performance. Within each level, the following three variables are collected for each target and star presented: whether the target or star was hit, the corresponding reaction time, and the duration of time each target or star was present on the screen.

**Laboratory-based inhibitory control tasks.** Consistent with prior work [39], inhibitory control was measured in-laboratory using the Anti-saccade Task, AX Continuous Performance Task (AXCPT), Stop Signal Delay Task (SST), and Flanker Task. Inhibitory control was operationalized using one variable from each task, including the percentage correct anti-saccade trials, AXCPT d-prime context, stop signal reaction time (SSRT), and the reaction time (RT) difference between congruent and incongruent flanker trials. For a full description of each task, please see S1 File.

**Clinical symptoms.** In addition to categorical diagnoses, dimensional levels of clinical symptoms were assessed. ADHD symptoms were assessed using the parent-report Conners Comprehensive Behavior Ratings Scale (CBRS) [40]. Total scores on the DSM-IV ADHD subscale were calculated. Additionally, consistent with prior work, [34,41] six items from the DSM-IV ADHD subscale were used to model ADHD in a bifactor model of clinical symptoms (see Statistical Analyses below). Irritability symptoms were assessed using the parent- and child-report Affective Reactivity Index (ARI) [42]. Parent-report ARI and child-report ARI Total scores were calculated. Anxiety symptoms were assessed using the parent- and child-report Screen for Anxiety Related Emotional Disorders (SCARED) [43]. Subscale scores were calculated for each of the five subscales (Generalized Anxiety, Panic, School Anxiety, Separation Anxiety, and Social Anxiety) separately for the parent- and child-report SCARED. Parent- and child-report SCARED Total scores were calculated as the sum across all items. For secondary analyses, we assessed depression using the parent and child-report Mood and Feelings Questionnaire (MFQ) [44]. Total scores on the MFQ were calculated for each participant.

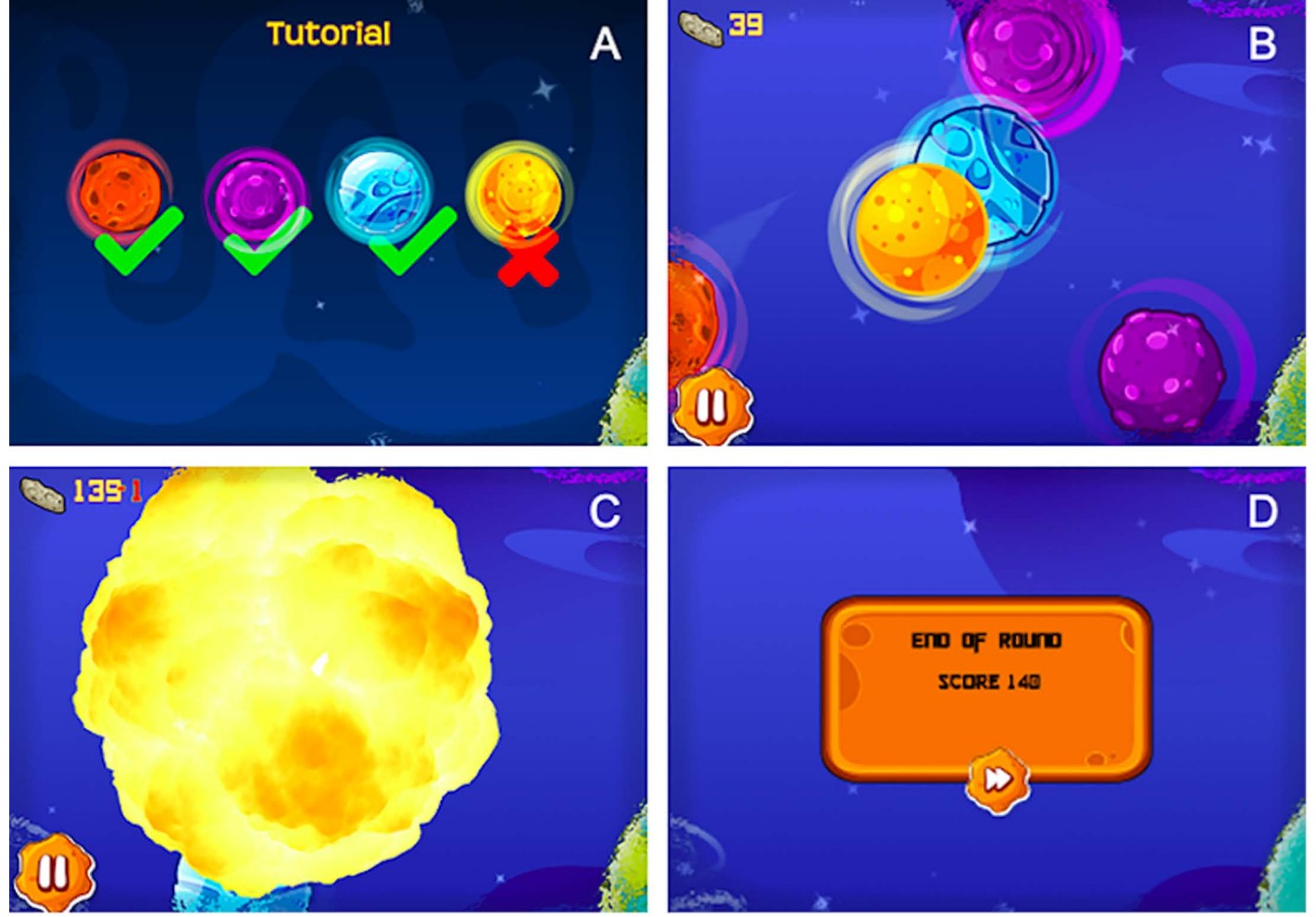

**Fig 3. Screenshots depicting CALM-IT gameplay.** (A) Tutorial screen with instructions, (B) In-level play depicting stars (yellow) and targets, (C) Explosion and negative point indicator feedback in response to swiping a star, (D) End of level cumulative score display.

**Neural response to cognitive conflict.** Participants underwent fMRI scanning while completing a modified version of the Eriksen Flanker Task [45], which consisted of four six-minute runs divided into three blocks. During this task, participants were shown five arrows presented side-by-side centered on the screen and instructed to press a button to indicate which direction the central arrow was pointing as quickly as possible. Trials consisting of all five arrows pointing in the same direction were categorized as congruent, while trials consisting of flanking arrows pointing in the *opposite* direction of the central arrow were categorized as incongruent. Participants received feedback between blocks based on their task performance in order to ensure a balance between the maintenance of accuracy with the maximization of errors [46].

Across four functional runs, 170 whole brain T*2 weighted echo-planer images were acquired using a 3T MR750 General Electric Scanner (Waukesha, Wisconsin, USA) and a 32-channel head coil (TR = 2000ms, TE = 25, flip angle = 60, field of view = 96x96, slices = 42/axial/3mm). For each participant, we also collected structural images using a magnetization-prepared rapid acquisition gradient echo (MPRAGE) sequence (TR = 7.66, TE = 3.42, flip angle = 7, field of view = 256x256, slices = 176/sagittal/1mm). All data were preprocessed and analyzed using Analysis of Functional Neuroimages (AFNI; version_23.1.08) [47] and Freesurfer [48] (for full details see S1 File).

## Statistical analyses

**Extracting mobile application-based measures of inhibitory control.** App-based inhibitory control behavior was operationalized using the variable d-prime from signal detection theory (SDT). CALM-IT behavior can be categorized in the following manner: correctly swiping at targets as a "correct hit", not swiping at targets as a "miss", incorrectly swiping at stars as a "false alarm," and not swiping stars as "correct rejection". d-prime is a measure of an individual's ability to discriminate signal from noise through the assessment of the degree of overlap between the standardized probability of hit trials and the standardized probability of false-alarm trials. A value of 0 corresponds with an inability to discriminate signal from noise; higher values correspond to better ability to discriminate signal from noise, indicating better inhibitory control performance. As such, CALM-IT d-prime assesses both the ability to successfully engage in target motor responses while simultaneously successfully engaging motor inhibition when required. Examination of either correct hit rate or false alarm rate alone would insufficiently capture our behavioral target. For example, participants who were overly cautious or distracted and withheld swipes for both targets and stars would inaccurately appear to have strong inhibitory control performance if false-alarm trials alone were examined. For each participant, d-prime was calculated as the difference between the standardized percentage of correct hits and the standardized percentage of false alarms. In order to allow for standardization of these probabilities using z-scores, extreme values for the hit rate and false-alarm rate (i.e., rates of 0 ($n = 12$, 2.51% of data) or 100 ($n = 1$, 0.21% of data) were adjusted following recommended practices [49].

**Temporal consistency of CALM-IT performance.** Test-retest reliability of CALM-IT task performance was analyzed in IBM SPSS Statistics (v28) using intraclass correlation coefficients (ICC), which capture both the degree of correlation and the agreement across measurements. We used a two-way mixed effects assessment of consistency (ICC(3,1)). Values of less than 0.40 indicate poor consistency, values of 0.40–0.70 indicate moderate consistency, valued of 0.70–0.90 indicate good consistency, and values greater than 0.90 indicate excellent consistency.

**Inhibitory control laboratory task performance.** Using measured variables from each of the four laboratory tasks, a latent factor of inhibitory control was estimated using confirmatory factor analyses conducted in Mplus (v8.5). Fit indices indicated good fit (CFI = 0.989, TLI = 0.967, RMSEA = 0.040, $CI_{90}$[0–0.129]) and all four indicators significantly loaded on the latent factor (all $ps < .01$, S1 File for more details on CFA analyses). Inhibitory control latent factor scores were extracted for each participant, with higher scores indicating better inhibitory control performance. Associations between inhibitory control latent factor scores and CALM-IT inhibitory control performance was examined using bivariate correlations. All analyses were repeated controlling for age.

**Neural response to cognitive conflict.** Analysis of Functional Neuroimages (AFNI; v_23.1.08) and Freesurfer were used for all imaging analyses. Seed coordinates for ROIs were selected using a Neurosynth term-based meta-analysis with the term "cognitive control". z-value maps (uniformity test for "cognitive control"), FDR-corrected to 0.01, were further thresholded to a z-value of 10. Spheres with a 6-mm radius (57 voxels) from peak coordinates were created for three ROIs: the anterior cingulate (ACC), supplementary motor area (SMA), and the bilateral inferior frontal gyrus (IFG). These ROIs were selected because of their central role in mediating attentional control and behavioral output in cognitive control tasks [13,50,51]. Percent signal change for each condition (congruent, incongruent) was extracted per participant using AFNI's 3dROIstat. For each of the three ROIs, we conducted an ANCOVA with Flanker condition as a within-subjects variable, and CALM-IT performance and age as between-subjects variables. The primary interaction of interest was the Flanker condition-by-CALM-IT performance interaction representing associations between brain activation during cognitive conflict and cognitive control performance of CALM-IT.

**Bifactor model of childhood psychopathology.** Consistent with prior work [39], shared and unique variances of irritability, anxiety, and ADHD symptoms were estimated using a bifactor model conducted in Mplus (v8.5), in which a general factor was estimated using scores on the five parent-report SCARED, six parent-report ARI items, and six CBRS items. An anxiety-specific latent factor, an irritability-specific latent factor, and an ADHD-specific latent factor were identified using the five parent-report SCARED, six parent-report ARI items, and six CBRS items, respectively. Model fit

indices indicated good model fit (CFI = 0.993, TLI = 0.991, RMSEA = 0.049, $CI_{90}$[0.029–0.066]; see S1 File for more details on CFA analyses). Factor scores on each of the four latent factors were extracted for each participant. Associations between clinical symptoms and CALM-IT inhibitory control performance was examined using bivariate correlations in IBM SPSS Statistics (v28). All analyses were repeated controlling for age.

## Results

### CALM-IT feasibility and engagement

Only five participants (2.5%) aborted CALM-IT prior to completion. Four participants (2%) completed CALM-IT, but their data was not collected due to technical issues. Of the remaining participants, eight participants were excluded from analyses of CALM-IT levels 1–5 and 12 participants were excluded from analyses of levels 6–10, due to technical difficulties impacting the number of targets and stars presented. For three participants (2.26%), Time 2 data were not collected, due to technical issues. Time 2 levels 1–5 data were excluded for six participants (4.62%) and Time 2 levels 6–10 data were excluded for four participants (3.11%), due to technical glitches impacting the number of targets and stars presented.

Demonstrating feasibility and acceptability, 94.79% of participants who were contacted to assess interest in completing CALM-IT agreed to do so. Participants completed 9.67 (SD = 1.12) out of 10 levels on average. Average performance metrics indicated strong engagement with CALM-IT, such that, on average, participants successfully hit 91.28% of targets (SD = 7.59) for Levels 1–5 and 78.03% of targets (SD = 8.73) for Levels 6–10, and successfully avoided hitting 79.80% of stars (SD = 17.71) for Levels 1–5 and 87.54% of stars (SD = 14.56) for Levels 6–10. Consistent with these findings, the average d-prime for levels 1–5 (M[SD] = 2.43[0.80]) and levels 6–10 (M[SD] = 2.13[0.56]) indicate that participants are able to discriminate signal (targets) from noise (stars).

### CALM-IT test-retest reliability

To assess temporal stability, we assessed test-retest reliability in the subset of participants who completed CALM-IT twice, approximately one week apart. Results indicated moderate reliability for levels 1–5 percentage of targets hit, ICC(3,1)=0.60 CI [0.47–0.70], percentage of stars hit, ICC(3,1)=0.50 CI [0.36–0.63], and CALM-IT d-prime, ICC(3,1)=0.50 CI [0.35–0.62], and for levels 6–10 percentage of targets hit, ICC(3,1)=0.61 CI [0.49–0.71], percentage of stars hit, ICC(3,1)=0.62 CI [0.49–0.72], and CALM-IT d-prime, ICC(3,1)=0.48 CI [0.33–0.61].

### CALM-IT convergent validity

**Associations with in-laboratory inhibitory control behavioral tasks.** Bivariate correlations between CALM-IT d-prime and inhibitory control latent factor scores indicated that, as predicted, higher CALM-IT d-prime scores were associated with better inhibitory control performance on canonical in-laboratory tasks for both levels 1–5, $r = .35$ $p < .001$, and levels 6–10, $r = 0.35$, $p < .001$. We repeated analyses controlling for age following examination of correlations with CALM-IT d-prime ($ps < .001$). After controlling for age, no associations between CALM-IT d-prime and inhibitory control latent factor scores survived (Levels 1–5 $b = 0.14$ $p = 0.128$, Levels 6–10 $b = 0.14$, $p = .133$). In both models, age remained a significant predictor of inhibitory control, with increasing age associated with better inhibitory control performance (all $ps < .001$).

**Associations with neural response to conflict.** The right and left IFG showed a condition-by-CALM-IT d-prime score interaction for levels 6–10 (left IFG: $F(1,48)=4.88$, $p = .032$; right IFG: $F(1,48)=7.68$, $p = .008$), but not levels 1–5 (all $ps > .05$), such that increased activation in the bilateral IFG in response to conflict was associated with better CALM-IT performance. No interaction emerged for the ACC/SMA for either metric (all $ps > .05$). A complementary whole-brain analysis did not yield any significant clusters at an appropriately stringent cluster correction (S1 File). All analyses controlled for age by including age as between-subjects variable.

## Associations with clinical phenotypes

Consistent with prior work [39], examination of associations of CALM-IT d-prime with factor scores derived from our bifactor model revealed associations with higher general psychopathology factor scores, but not any of the symptom-specific factors (S1 File). Associations between general psychopathology factor scores remained significantly associated with both CALM-IT levels 1–5 d-prime, $b = -0.16$ $p = 0.047$, and levels 6–10 d-prime, $b = -0.22$ $p = 0.006$, even after controlling for age.

When examining associations with self- and parent-report symptom assessments total scores, associations emerged between CALM-IT d-prime and higher levels of irritability (parent-report with levels 1–5 d-prime: $r(165) = -0.21$ $p = .006$ and levels 6–10: $r(162) = -0.19$ $p = .018$; child-report with levels 1–5 d-prime: $r(164) = -0.19$ $p = .013$ and levels 6–10: $r(161) = -0.18$ $p = .025$) and ADHD symptoms (levels 1–5 d-prime: $r(163) = -0.20$ $p = .010$ and levels 6–10: $r(162) = -0.30$ $p < .001$), but not anxiety symptoms. Last, exploratory analyses examining associations with subtypes of ADHD and depression found associations with higher hyperactive impulsive ADHD scores CALM-IT performance across levels 1–5 ($r(163) = -0.27$ $p < .001$) and levels 6–10 ($r(161) = -0.33$ $p < .001$) and associations with higher inattentive ADHD scores for CALM-IT performance on levels 6–10 ($r(161) = -0.22$ $p = .006$). No significant associations emerged for measures of depression (S1 File). After controlling for age, all associations between self- and parent-report symptoms with CALM-IT performance were no longer significant.

## Discussion

In the current study we leveraged mobile technology to develop a novel, gamified, and easily disseminable tool, "CALM-IT", to experimentally assess inhibitory control behaviors in vivo. When following our roadmap for the development of novel tools, four main findings arose. First, the inhibitory control mobile app was accessible, feasible, and engaging. Second, CALM-IT performance was moderately reliable over time. Third, performance related to both a latent variable of inhibitory control from standardized in-clinic tasks and neural activation. However, only associations with neural activation to conflict in the IFG remained significant after controlling for age. Finally, CALM-IT performance was related to increased levels of co-occurring anxiety, irritability, and ADHD symptoms.

We applied knowledge from two canonical inhibitory control laboratory-based tasks, Go/No-Go and Stop Signal Delay task, to inform CALM-IT's methodological design. In doing so, we translated these inhibitory control tasks into developmentally appropriate and engaging levels of gameplay, whereby participants were presented with go-stimuli (i.e., space objects including comets and asteroids) and no-go/stop stimuli (i.e., stars or comets that turn into stars).

Next, we demonstrated feasibility and engagement with CALM-IT. Youth and families overwhelmingly exhibited interest in participating in the protocol and once enrolled, participants had high completion rates and performance accuracy confirming that the task design aligns with the developmental capacities and skills of clinical pediatric samples. Next, we demonstrated moderate test-retest reliability of inhibitory control behavior as measured by CALM-IT, across two time points, approximately 1 week apart. These findings provide confidence in the stability of inhibitory control behavior as measured by CALM-IT. Importantly, moderate levels of test-retest reliability suggest that CALM-IT may be leveraged to gain repeated measures of inhibitory control behavior across time to capture intra-individual variation. As such, CALM-IT could be integrated into ecological momentary assessment protocols to provide repeated experimental measures of inhibitory control across time and thus allowing for the investigation of temporal relationships between mood, clinical symptoms, and other clinically relevant behaviors (i.e., sleep) through within-subject repeated measures of these behaviors (e.g., EMA paired with CALM-IT).

To assess convergent validity of behaviors measured by CALM-IT, we examined associations with a latent factor of inhibitory control estimated across four canonical in-laboratory tasks. While associations emerged between CALM-IT inhibitory control behavior and inhibitory control latent factor scores, this association did not remain after controlling for age. Given that our current sample encompasses a developmental period characterized by rapid development of inhibitory

control, [17] it is unsurprising that a strong association between age and inhibitory control emerged as measured both by in-laboratory tasks and CALM-IT. Developmental approaches toward validating these novel mobile applications are of vital importance to ensure robust measures of inhibitory control that match the capacity for inhibitory control across developmental stages.

Neural correlates of inhibitory control were assessed in a subset of participants who underwent functional magnetic resonance imaging (fMRI) while completing the Eriksen Flanker Task. Region of interest analyses identified associations between activation in the bilateral IFG and CALM-IT performance for the last five levels thus linking biological correlates of inhibitory control to CALM-IT performance. The IFG is consistently demonstrated to be particularly important for inhibitory control processing [13] and atypical activation in the IFG during inhibitory control has been linked to elevated irritability [46,52]. Of note, the observed association was with levels 6–10, but not levels 1–5 of CALM-IT. Levels 6–10 were modeled in part on the Stop Signal Delay paradigm and, therefore, include go-signals that become stop-signals. This introduces increased difficulty and more variability in performance on those levels. Within this sample encompassing late childhood through adolescence, this increased difficulty may have better captured developmental variance in inhibitory control and activation of the underlying neural circuitry, such as the IFG.

Finally, we identified clinical relevance through associations with measures of anxiety, irritability, and ADHD symptoms and found that impaired inhibitory control, as measured by CALM-IT, was associated with higher levels of the general latent factor for our bifactor model of clinical symptoms (i.e., shared or comorbid psychopathology), even after controlling for age. These findings are consistent with a growing body of research identifying inhibitory control as a risk factor for psychopathology broadly and comorbidity across clinical symptoms [39,53]. As such, easily accessible and disseminatable tools for experimentally assessing inhibitory control in vivo, such as CALM-IT, have significant clinical implications for childhood psychopathology broadly.

The current application of our proposed framework to the development of CALM-IT has several limitations. First, while overall engagement with CALM-IT was strong, we faced technical issues that resulted in loss of data. These difficulties highlight the importance of strong technical partnerships and flexible/user friendly technical requirements for both the running of the application, as well as data storage and upload. Next, there was variability across participants in the time elapsed between CALM-IT gameplay and measures of in-laboratory inhibitory control behavior and neural response to cognitive conflict. Future work examining behaviors measured consistently close in time may find more robust associations. Finally, the current sample was enriched for anxiety, irritability, and ADHD symptoms and, as such, the analyses examining clinical relevance focused on these symptom domains. However, other clinical symptoms may be relevant and should be considered in future work (i.e., depressive and conduct disorder symptoms).

Validation of this neuroscience-informed mobile application represents a critical first step forward in bridging the gap between granular, mechanism-driven basic science research and community-based assessment and treatment of childhood psychopathology. There is a profound need to use neuroscience-based assessments to inform treatment decisions. However, such assessments are not accessible to community providers. The goal of this work is to bridge that gap by creating and facilitating the usage of an accessible and engaging novel mobile application that probes a validated behavioral construct. The present work lays the groundwork for an important line of future work that could provide researchers and clinicians a multifaceted tool to measure multiple aspects of clinically relevant behaviors such as inhibitory control in an engaging and accessible manner. Finally, advances in digital mental health methodology, like those outlined in this work, represent an important step forward in increasing representative samples and accessibility of clinical research. By leveraging mobile technology, we aim to bridge the gap between precise mechanism-driven basic science and the real-world clinical outcomes these mechanisms are hypothesized to underlie. This work represents an important first step, and future work is needed examine the potential for targeted mobile applications such as CALM-IT to predict symptom time courses and treatment outcomes. We hope that researchers will consider the roadmap outlined in this work when pursuing these goals.

## Supporting information

**S1 File. Supplementary file containing all supplementary text, tables, and figures.**
(PDF)

## Author contributions

**Conceptualization:** Elise Cardinale, Simone P Haller, Reut Naim, Daniel S Pine, Melissa A Brotman.

**Data curation:** Elise Cardinale, Jennifer M Meigs, Simone P Haller, Kenny Fling, Urmi Pandya, Olivia Siegal, Anjali Poe, Shannon Shaughnessy, Christian Zapp, Jessica L Bezek, Parmis Khosravi, Lauren M Henry.

**Formal analysis:** Elise Cardinale, Simone P Haller.

**Methodology:** Elise Cardinale, Simone P Haller, Kyunghun Lee, David C Jangraw, Katharina Kircanski, Reut Naim, Daniel S Pine, Melissa A Brotman.

**Resources:** Ellen Leibenluft, Daniel S Pine, Melissa A Brotman.

**Supervision:** Elise Cardinale, Katharina Kircanski, Ellen Leibenluft, Daniel S Pine, Melissa A Brotman.

**Writing – original draft:** Elise Cardinale, Jennifer M Meigs, Simone P Haller, Ramaris German, Melissa A Brotman.

**Writing – review & editing:** Elise Cardinale, Jennifer M Meigs, Simone P Haller, Kenny Fling, Urmi Pandya, Olivia Siegal, Anjali Poe, Shannon Shaughnessy, Christian Zapp, Jessica L Bezek, Kyunghun Lee, Parmis Khosravi, Ramaris German, David C Jangraw, Lauren M Henry, Meghan E Byrne, Katharina Kircanski, Ellen Leibenluft, Reut Naim, Daniel S Pine, Melissa A Brotman.

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
