## [Decision Letter · Decision Letter 0]

10 Jan 2025

PONE-D-24-31796Leveraging technology to probe mechanisms of psychopathology: A proof of concept study of inhibitory controlPLOS ONE

Dear Dr. Cardinale,

Thank you for submitting your manuscript to PLOS ONE. After careful consideration, we feel that it has merit but does not fully meet PLOS ONE’s publication criteria as it currently stands. Therefore, we invite you to submit a revised version of the manuscript that addresses the points raised during the review process.

We look forward to receiving your revised manuscript.

Kind regards,

Rajneesh Choubisa

Academic Editor

PLOS ONE

Journal Requirements:

This research was supported by the NIMH Intramural Research Program (ZIAMH002781), conducted under NIH Clinical Study Protocols 15-M-0182 (ClinicalTrials.gov identifier: NCT02531893), 02-M-0021 (NCT00025935), 00-M-0198 (NCT00006177), and 01-M-0192 (NCT00018057).  

This research was supported by the NIMH Intramural Research Program

(ZIAMH002781), conducted under NIH Clinical Study Protocols 15-M-0182

(ClinicalTrials.gov identifier: NCT02531893), 02-M-0021 (NCT00025935), 00-M-0198

(NCT00006177), and 01-M-0192 (NCT00018057).

This research was supported by the NIMH Intramural Research Program (ZIAMH002781), conducted under NIH Clinical Study Protocols 15-M-0182 (ClinicalTrials.gov identifier: NCT02531893), 02-M-0021 (NCT00025935), 00-M-0198 (NCT00006177), and 01-M-0192 (NCT00018057).

5. Your abstract cannot contain citations. Please only include citations in the body text of the manuscript, and ensure that they remain in ascending numerical order on first mention.

6. We note that Figures 1, 2, and 3 in your submission contain copyrighted images. All PLOS content is published under the Creative Commons Attribution License (CC BY 4.0), which means that the manuscript, images, and Supporting Information files will be freely available online, and any third party is permitted to access, download, copy, distribute, and use these materials in any way, even commercially, with proper attribution. For more information, see our copyright guidelines: http://journals.plos.org/plosone/s/licenses-and-copyright.

a. You may seek permission from the original copyright holder of Figures 1, 2, and 3 to publish the content specifically under the CC BY 4.0 license. 

Additional Editor Comments :

Dear Authors,

I have now received reviews from all the reviewers and their comments are not very positive for the manuscript submitted by the authors. Considering my own reading and the comments provided, I invite the authors to provide a solid rebuttal to the reviewers and submit a revised version of the manuscript which should be better version of the original version. Please go through the comments of each of the reviewers and provide a line by line rebuttal so that it can taken up for further consideration by the journal. I am also copy pasting the comments from both the reviewers for your ready reckoning.

Reviewer 1: Reject

The submission “Leveraging technology to probe mechanisms of psychopathology: A proof of concept study of inhibitory control “ reports several results from a newly developed game that is proposed to measures inhibitory control. The authors report data on the reliability, convergent validity with laboratory-based measures, associations with mental health symptoms (subclinical ADHD) and fMRI. The authors tested a large sample of adolescents, mostly diagnosed with ADHD or mood/anxiety disorder, or healthy controls.

Overall, the submission is very comprehensive and contributes a lot of exciting results. My concern with the submission is that the research dilutes the precision of inhibitory control tasks, and their new app presents a unitary approach that claims to assess inhibitory control. It does not further inform about mechanisms, though, at least in the current shape of the submission – more information about the correlations with individual laboratory tasks, and much more details about the tasks themselves are needed. It should be discussed and made transparent to future users of the app that their measurement is not precisely characterized as inhibitory control in other lab paradigms. Furthermore, the presentation is not very critical and does not (yet) provide a clear description of the scientific contribution.

- Was the study preregistered? It says “registered report” under article type, I assume this is a mistake and should be an original report, because I could not find any mention of the stage I submission or the preregistration of their data analyses.

- I could not review whether all data are available, but the authors stated that they would publish all data on OpenNeuro. In my view this is perfectly fine and it will certainly be a great asset for future meta-analytic studies.

- More precision in the introduction regarding what the innovation of the app is required. What aspect of inhibitory control is measured, and what is the benefit of using the app instead of a brief but precise laboratory assessment, under much better controlled conditions?

- The description of the task is missing a lot of detail to understand what participants did. How long were stimuli presented? Which stimuli were presented? In the SST variant, which delays were randomly presented and how often? Are participants instructed not to wait on the task? The consensus guide on SST can be used to check whether all relevant data are reported (Verbruggen et al., 2019, eLife).

- The authors reported d-prime, collected from their app, as a measure of inhibitory control. I would like to reflect whether inhibitory control is best defined by taking into considerations all hits, misses, across go-/no-go and stop-signal conditions. It is not clear what their task measures and what the outcome reflects.

- Related to the point above, I would like to see the separate correlations with SSRT, false alarms, congruency effect in the Flanker task, and so on, rather than a composite latent score. This will help to better elucidate the cognitive mechanism that is measured.

- The association of age and measures from the tasks is mentioned in the discussion, please also include the results.

- Roadmap: I wonder whether the authors would agree, in retrospect, to include more early piloting and technical testing (given the reported loss of data due to technical issues). I also wondered whether there was any patient involvement in the development of the app. How does the reliability, and validity, inform the further steps?

- A commercial partner was involved in the design of the app. Please state in the conflicts of interest whether the partner was involved in the study design, analysis, or decision to submit for publication.

- Are the reported reliabilities and validities sufficient to follow up with a broader adaptation of the app? And what is the actual use case? I am not convinced that the app provides a solid basis for treatment decisions in the current state

- The effects of the employed gamification elements on inhibitory control assessment should be discussed. There are several studies that reported positive and negative effects on engagement but also on performance, e.g.:

Friehs, M. A., Dechant, M., Vedress, S., Frings, C., & Mandryk, R. L. (2020). Effective gamification of the stop-signal task: two controlled laboratory experiments. JMIR Serious Games, 8(3), e17810.

Schroeder, P. A., Lohmann, J., & Ninaus, M. (2021). Preserved inhibitory control deficits of overweight participants in a gamified stop-signal task: experimental study of validity. JMIR Serious Games, 9(1), e25063.

Lumsden, J., Skinner, A., Coyle, D., Lawrence, N., & Munafo, M. (2017). Attrition from web-based cognitive testing: a repeated measures comparison of gamification techniques. Journal of medical Internet research, 19(11), e395.

Reviewer 2: Major Revision

Major comments:

- Second hypothesis: Why is accuracy conceptualized as an index of task validity rather than an index of the capacities and skills of users?

- CALM-IT: how do stard become a prepotent signal as in traditional go/no-go tasks, where the inhibition is inhibition of a prepotent response?

- More information is needed on statistical tests: please list all software and versions that were used for each analysis. please, list all relevant assumptions and how those were tested. ICCs are also insufficiently described, e.g. was absolute agreement assessed?

More detail should also be provided on the CFA. Provide references for the cutoffs used.

- it is unclear why the results state "). After controlling for age, all associations between self- and parent-report symptoms with CALM-IT performance were no longer significant." but the discussion and abstract treats this findings as remaining significant after accounting for age.

- it does not appear that - given the novelty of the research question - region of interest analyses are warranted. ROI analyses introduce pre-selection bias and have less generalizability and limited scope. there is not enough precedent to justify a ROI approach.

I leave it to the discretion of authors to work on the revision and submit the same as per allocated timeline or take any appropriate decision on this manuscript. However, I would be pleased if they can work on a revision and submit a revised version for reconsideration by the journal.

Best Wishes,

Academic Editor

Reviewers' comments:

Reviewer's Responses to Questions

**Comments to the Author**

1. Does the manuscript adhere to the experimental procedures and analyses described in the Registered Report Protocol?

If the manuscript reports any deviations from the planned experimental procedures and analyses, those must be reasonable and adequately justified.

Reviewer #1: No

Reviewer #2: Yes

2. If the manuscript reports exploratory analyses or experimental procedures not outlined in the original Registered Report Protocol, are these reasonable, justified and methodologically sound?

A Registered Report may include valid exploratory analyses not previously outlined in the Registered Report Protocol, as long as they are described as such.

Reviewer #1: Partly

Reviewer #2: Yes

3. Are the conclusions supported by the data and do they address the research question presented in the Registered Report Protocol?

The manuscript must describe a technically sound piece of scientific research with data that supports the conclusions. The conclusions must be drawn appropriately based on the research question(s) outlined in the Registered Report Protocol and on the data presented.

Reviewer #1: No

Reviewer #2: Partly

4. Have the authors made all data underlying the findings in their manuscript fully available?

Reviewer #1: No

Reviewer #2: Yes

5. Is the manuscript presented in an intelligible fashion and written in standard English?

Reviewer #1: Yes

Reviewer #2: Yes

6. Review Comments to the Author

Please use the space provided to explain your answers to the questions above. (Please upload your review as an attachment if it exceeds 20,000 characters)

Reviewer #1: The submission “Leveraging technology to probe mechanisms of psychopathology: A proof of concept study of inhibitory control “ reports several results from a newly developed game that is proposed to measures inhibitory control. The authors report data on the reliability, convergent validity with laboratory-based measures, associations with mental health symptoms (subclinical ADHD) and fMRI. The authors tested a large sample of adolescents, mostly diagnosed with ADHD or mood/anxiety disorder, or healthy controls.

Overall, the submission is very comprehensive and contributes a lot of exciting results. My concern with the submission is that the research dilutes the precision of inhibitory control tasks, and their new app presents a unitary approach that claims to assess inhibitory control. It does not further inform about mechanisms, though, at least in the current shape of the submission – more information about the correlations with individual laboratory tasks, and much more details about the tasks themselves are needed. It should be discussed and made transparent to future users of the app that their measurement is not precisely characterized as inhibitory control in other lab paradigms. Furthermore, the presentation is not very critical and does not (yet) provide a clear description of the scientific contribution.

- Was the study preregistered? It says “registered report” under article type, I assume this is a mistake and should be an original report, because I could not find any mention of the stage I submission or the preregistration of their data analyses.

- I could not review whether all data are available, but the authors stated that they would publish all data on OpenNeuro. In my view this is perfectly fine and it will certainly be a great asset for future meta-analytic studies.

- More precision in the introduction regarding what the innovation of the app is required. What aspect of inhibitory control is measured, and what is the benefit of using the app instead of a brief but precise laboratory assessment, under much better controlled conditions?

- The description of the task is missing a lot of detail to understand what participants did. How long were stimuli presented? Which stimuli were presented? In the SST variant, which delays were randomly presented and how often? Are participants instructed not to wait on the task? The consensus guide on SST can be used to check whether all relevant data are reported (Verbruggen et al., 2019, eLife).

- The authors reported d-prime, collected from their app, as a measure of inhibitory control. I would like to reflect whether inhibitory control is best defined by taking into considerations all hits, misses, across go-/no-go and stop-signal conditions. It is not clear what their task measures and what the outcome reflects.

- Related to the point above, I would like to see the separate correlations with SSRT, false alarms, congruency effect in the Flanker task, and so on, rather than a composite latent score. This will help to better elucidate the cognitive mechanism that is measured.

- The association of age and measures from the tasks is mentioned in the discussion, please also include the results.

- Roadmap: I wonder whether the authors would agree, in retrospect, to include more early piloting and technical testing (given the reported loss of data due to technical issues). I also wondered whether there was any patient involvement in the development of the app. How does the reliability, and validity, inform the further steps?

- A commercial partner was involved in the design of the app. Please state in the conflicts of interest whether the partner was involved in the study design, analysis, or decision to submit for publication.

- Are the reported reliabilities and validities sufficient to follow up with a broader adaptation of the app? And what is the actual use case? I am not convinced that the app provides a solid basis for treatment decisions in the current state

- The effects of the employed gamification elements on inhibitory control assessment should be discussed. There are several studies that reported positive and negative effects on engagement but also on performance, e.g.:

Friehs, M. A., Dechant, M., Vedress, S., Frings, C., & Mandryk, R. L. (2020). Effective gamification of the stop-signal task: two controlled laboratory experiments. JMIR Serious Games, 8(3), e17810.

Schroeder, P. A., Lohmann, J., & Ninaus, M. (2021). Preserved inhibitory control deficits of overweight participants in a gamified stop-signal task: experimental study of validity. JMIR Serious Games, 9(1), e25063.

Lumsden, J., Skinner, A., Coyle, D., Lawrence, N., & Munafo, M. (2017). Attrition from web-based cognitive testing: a repeated measures comparison of gamification techniques. Journal of medical Internet research, 19(11), e395.

Reviewer #2: I appreciate the opportunity to review this manuscript. The writing is clear and organized, the aims are clinically and practically relevant.

Major comments:

- Second hypothesis: Why is accuracy conceptualized as an index of task validity rather than an index of the capacities and skills of users?

- CALM-IT: how do stard become a prepotent signal as in traditional go/no-go tasks, where the inhibition is inhibition of a prepotent response?

- More information is needed on statistical tests: please list all software and versions that were used for each analysis. please, list all relevant assumptions and how those were tested. ICCs are also insufficiently described, e.g. was absolute agreement assessed?

More detail should also be provided on the CFA. Provide references for the cutoffs used.

- it is unclear why the results state "). After controlling for age, all associations between self- and parent-report symptoms with CALM-IT performance were no longer significant." but the discussion and abstract treats this findings as remaining significant after accounting for age.

- it does not appear that - given the novelty of the research question - region of interest analyses are warranted. ROI analyses introduce pre-selection bias and have less generalizability and limited scope. there is not enough precedent to justify a ROI approach.

7. PLOS authors have the option to publish the peer review history of their article (what does this mean? ). If published, this will include your full peer review and any attached files.

**Do you want your identity to be public for this peer review?** For information about this choice, including consent withdrawal, please see our Privacy Policy .

Reviewer #1: **Yes: ** Philipp A. Schroeder

Reviewer #2: No

---

## [Author Response · Author response to Decision Letter 1]

17 Jan 2025

Please see Response to Reviewers attachment as the response contains tables and formatting that do not appear well when entered into the text box.

---

## [Editor Report · Decision Letter 1]

27 Jan 2025

Leveraging technology to probe mechanisms of psychopathology: A proof of concept study of inhibitory control

PONE-D-24-31796R1

Dear Dr. Cardinale,

We’re pleased to inform you that your manuscript has been judged scientifically suitable for publication and will be formally accepted for publication once it meets all outstanding technical requirements.

Kind regards,

Rajneesh Choubisa, Ph.D

Academic Editor

PLOS ONE

Additional Editor Comments (optional):

Dear Authors,

Thank you for providing rebuttal and incorporating the requested details in the revised version of the manuscript. The revision is now clarifies the intent of the diagnostic tool CALM-IT. I know that authors are not sure at this time whether this would be validated patented tool usable for pediatricians and other experts in the future as they provide it as a mere roadmap. Nevertheless, I also believe that authors should rope in a computer scientist cum app developer to further validate and upload the CALM-IT tool on Google Playstore/Apple store for furthering their assessment platform's accessibility from a utilitarian perspective. I would also recommend that authors should fix the errors (at least the first two to make it BIDS compliant) in the uploaded dataset on OpenNeuro (ds005166) because it will now be accessible to the general public (link provided now) for analysis and replication, whenever accessed. As a researcher, I am highly curious in exploring the levels 1-10 of the four test scenarios (figure 3) and how they have been quantified to check for CFA. The authors state that the number of successful trials on each of the tasks were considered and subsequently quantified, which was inadequately detailed and elaborated.

Considering the conclusion, I wish that authors should go a step ahead to develop, test, and validate the CALM-IT platform and make it more inclusive for all sort of samples in near future. I believe the authors also have an opportunity to get it patented later. Adding more snapshots of the CALM-IT would make more sense.

Best Wishes,

Academic Editor
---

## [Editor Report · Acceptance letter]

PONE-D-24-31796R1

PLOS ONE

Dear Dr. Cardinale,

I'm pleased to inform you that your manuscript has been deemed suitable for publication in PLOS ONE. Congratulations! Your manuscript is now being handed over to our production team.

Kind regards,

on behalf of

Dr. Rajneesh Choubisa

Academic Editor

PLOS ONE